# Guided Wave Transducer for the Locating Defect of the Steel Pipe Based on the Weidemann Effect

**Jin Xu [1], Guang Chen [2], Jiang Xu [2],\* and Qing Zhang [3]**

[1] Colleague of Electronic and Information Engineering, Nanjing University of Aeronautics and Astronautics, Nanjing 211106, China; xujin95@nuaa.edu.cn
[2] School of Mechanical Science and Engineering, Huazhong University of Science and Technology, Wuhan 430074, China; m201670564@hust.edu.cn
[3] Colleague of Automation Engineering, Nanjing University of Aeronautics and Astronautics, Nanjing 211106, China; zhangqing@nuaa.edu.cn
\* Correspondence: jiangxu@mail.hust.edu.cn; Tel.: +86-2787559332

**Abstract:** The electromagnetic guided wave transducer has been widely used in pipeline detection in recent years due to its non-contact energy conversion characteristics. Based on the Weidemann effect, an electromagnetic guided wave transducer that can realize the locating defect of the steel pipe was provided. Firstly, the principle of the transducer was analyzed based on the Weidemann effect. The basic structure of the transducer and the basic functions of each part were given. Secondly, the key structural parameters of the transducer were studied. Based on the size of the magnets and the coils, a protype electromagnetic guided wave transducer based on Wiedemann effect was developed. Finally, the experiments were carried out on the steel pipe with a defect using the developed transducer. The results show that the transducer can actuate and receive the T(0,1) and T(1,1) modes in the steel pipe. The axial positioning of the defect is located by moving the transducer axially. The circumferential positioning of the defect is located by rotating the transducer. Additionally, missed detection can be effectively avoided by rotating the transducer.

**Keywords:** electromagnetic guided wave transducer; torsional mode; defect location; Weidemann effect; pipe

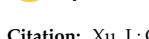



## 1. Introduction

Ultrasonic guided wave technology uses elastic waves propagating in a waveguide to detect defects. With a single point of excitation, it can realize long-distance detection, unreachable area detection, and high detection efficiency. The technology has been widely used in pipeline detection [1–9]. In order to actuate and receive guided waves in the pipeline, the common types of guided wave transducers are used, including piezoelectric, Lorentz force, and magnetostrictive [10,11]. The piezoelectric guided wave transducer is mainly based on the piezoelectric effect of the material [12]. The efficiency of energy conversion is greatly affected by the coupling state, and the surface of the component needs to be treated. The Lorentz force guided wave transducer [13] is a non-contact transducer. No surface treatment of the component is required. However, the transducer has low energy conversion efficiency, is greatly affected by lift-off, and is also affected by magnetostrictive force in ferromagnetic materials. The energy conversion of magnetostrictive guided wave transducer is mainly based on the magnetostrictive effect of the material [14,15]. Common magnetostrictive effects include the Joule effect and Wiedemann effect. The ferromagnetic material deforms under the static and dynamic magnetic field, which causes the particle in the pipe to vibrate, forming an elastic wave. Based on the Joule effect, the Lamb wave in the plate and the longitudinal mode guided wave in the cylinder are studied. Based on the Wiedemann effect, the shear horizontal wave in the plate and the torsional mode guided wave in the cylinder are studied. When the magnetostrictive guided wave transducer

uses high saturation magnetostrictive material as the transducer element, it is a contact transducer and requires a coupling agent to transmit the guided wave vibration [16]. The energy conversion efficiency is high, but the cost is high and the surface treatment of the component is required. When the energy conversion process of the magnetostrictive guided wave transducer directly occurs in the ferromagnetic material, it is a non-contact transducer and can only detect the ferromagnetic material but does not require surface treatment of the member. In addition, although the Lorentz force still exists in the ferromagnetic material, the guided wave energy excited by the magnetostrictive effect is much greater than the contribution of the Lorentz force [17].

Kwun [18] firstly developed a pipe magnetostrictive torsional guided wave transducer, which was composed of a nickel strip and a solenoid coil. However, the pre-magnetization state of the nickel strip is unstable and easy to demagnetize, which is not conducive to long-term monitoring. It is necessary to meet the conditions of equal dynamic and static magnetic field strength to suppress other modes, which is not conducive to on-site detection. Cho [19] used permanent magnets, coils, and iron-cobalt tape to excite a 2 MHz torsional guided wave in an aluminum tube and applied it to the detection of small defects. Cho's research results pointed out that the signal-to-noise ratio of the detection signal at high frequency is high. The coil has better frequency selection characteristics, but the high-frequency signal has a tail phenomenon and high-order torsional mode guided waves. Kim [20] eliminated signal smearing by impedance matching the sensor installation area and non-installation area. Kim [21] and Kwon [22] suppressed high-order torsional mode guided waves through two excitation transducers or constricted waveguide units, respectively. Liu [23] changed the turn-back coil in the transducer to a solenoid array coil, which not only retains the frequency selection characteristics of the sensor but also takes advantage of the strong excitation energy of the solenoid coil to enhance the torsional mode guided wave vibration.

Based on the Wiedemann effect, the torsional mode guided waves can be excited in the steel tube while avoiding the influence of the Lorentz force. However, the current research on the torsional mode guided wave transducer based on the Wiedemann effect mainly focuses on the contact transducer using magnetostrictive patches, which will cause surface damage if the steel pipe needs to be polished. There is a lack of a non-contact magnetostrictive torsion mode guided wave transducer. Based on the Weidemann effect, a noncontact guided wave transducer can actuate and receive the T(0,1) and T(1,1) modes in the steel pipe that is given in this paper. The transducer can locate the axial position and the circumferential position of the defect in the steel pipe.

## 2. Principle of the Transducer

The Wiedemann effect refers to the shear deformation of ferromagnetic materials under the action of an orthogonal dynamic and static magnetic field. In order to design a guided wave transducer based on the Wiedemann effect, it is necessary to study the direction of the excitation force formed by the Wiedemann effect in the steel pipe. By arranging the static and dynamic magnetic field to be distributed in the pipeline, the guided waves dominated by the desired mode will be excited [24]. Usually, the static magnetic field in ferromagnetic components is much larger than the dynamic magnetic field. Under the action of the dynamic and static magnetic field, the magnetostrictive constitutive relationship of ferromagnetic materials can be expressed as a linear piezomagnetic coupling model [25]:

$$\varepsilon = s^H \sigma + \psi^T H$$
$$B = \psi \sigma + \mu^\sigma H \tag{1}$$

where $\sigma$ and $\varepsilon$ represent the stress and strain matrix, respectively; $B$ and $H$ represent the magnetic induction intensity and magnetic field intensity matrix, respectively; $s^H$ represents the compliance matrix under constant magnetic field; $\mu^\sigma$ represents the magnetic permeability matrix under constant stress; $\psi$ represents the piezomagnetic coupling coefficient matrix.

Equation (1) can be decomposed into static and dynamic parts. Only the dynamic part needs to be considered when analyzing the guided wave transduction process.

$$\varepsilon_D = s^H \sigma_D + d^T H_D$$
$$B_D = d\sigma_D + \mu^\sigma H_D \tag{2}$$

where the subscript $D$ represents the dynamic component; $d$ represents the piezomagnetic coupling coefficient matrix related to the magnetization direction of the static magnetic field, which has directivity.

There are circumferential and axial magnetostrictive excitation forces in the steel pipe [24]:

$$f_{\theta^{MS}} = -\frac{3c_{44}^H \varepsilon(H_{0\theta})}{H_{0\theta}} \frac{\partial H_{Dz}}{\partial z} \tag{3}$$

$$f_{z^{MS}} = -\frac{3c_{44}^H \varepsilon(H_{0\theta})}{H_{0\theta}} \frac{\partial H_{Dz}}{\partial \theta} \tag{4}$$

where $c_{44}^H$ represents the elements in the stiffness matrix, $\varepsilon(H)$ represents the magnetostriction curve of the material; $H_{0\theta}$ represents the value of the circumferential static magnetic field, $H_{Dz}$ represents the axial dynamic magnetic field.

In the steel pipe, Equation (3) controls guided waves propagating in the axial direction that are circumferentially polarized, and Equation (4) controls guided waves propagating in the circumferential direction that are axially polarized. Because only axial propagation (torsional modes) is considered, the influence of Equation (4) is also ignored. In order to excite the torsional mode guided wave, it is necessary to provide an orthogonal magnetic field to generate a force in the circumferential direction. The static magnetic field, the dynamic magnetic field, and the force distribution on the surface of the steel pipe is shown in Figure 1.

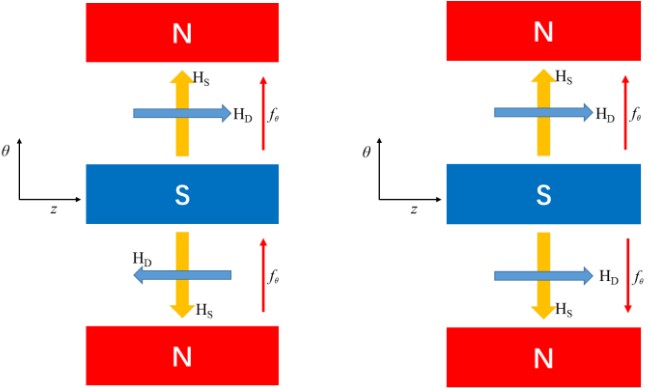

(**a**) For generating T(0,1) mode.    (**b**) For generating T(1,1) mode.

**Figure 1.** Schematic diagram of the static magnetic field and the dynamic magnetic field distribution on the surface of the steel pipe for generating T(0,1) mode and T(1,1) mode guided waves.

For the receiving process, the dynamic coil induces changes in the circumferential magnetic field. The additional magnetic field caused by guided wave vibration can be obtained as:

$$B_{D\theta} = \frac{3\varepsilon(H_{0z})\sigma_{D4}}{H_{0z}} \tag{5}$$

In summary, the combination of the circumferential static magnetic field and the axial dynamic magnetic field or the orthogonal magnetic field combination of the axial static magnetic field and the circumferential dynamic magnetic field can excite axially propagating and circumferentially polarized guided waves in the steel pipe. Furthermore, the circumferential magnetostrictive excitation force formed in the steel pipe and the

induced magnetic field change have similar expressions. As a non-contact transducer, the coil used to generate the dynamic magnetic field and the permanent magnet of the static magnetic field need to consider the influence of the lift-off distance. These theories are covered in the two published papers [26,27], and we can learn from them in the follow-up transducer characteristic parameter research.

## 3. The Design of the Transducer

### 3.1. The Structure of the Transducer

In order to excite the torsional mode guided waves in the steel pipe, it is necessary to provide a circumferential excitation force. The circumferential force can be generated by designing two orthogonal magnetic fields. There are two combinations. The first is the circumferential static magnetic field with the axial dynamic magnetic field. The second is the axial static magnetic field with the circumferential dynamic magnetic field. However, it is difficult to achieve a complete circumferential magnetic field for the circumferential structure. Therefore, we divided the circumferential direction into multiple equal sections. Permanent magnets were used to generate local static magnetization in the circumferential direction. Meander coils were used to generate local axial dynamic magnetization. The circumferential direction coverage was realized by the combination of multiple permanent magnets and meander coils.

The arrangement of permanent magnets and meander-line coils of the transducer is shown in Figure 2. Several permanent magnets and meander-line coils were arranged at equal intervals in the circumferential direction of the steel pipe. By arranging a specific number of permanent magnets and meander-line coils and then controlling the current direction of the meander-line coils, the specific torsional mode guided waves can be generated in the pipe. For example, in order to generate T(0,1) and T(1,1) mode guided waves, only two magnets and two meander-line coils can be employed, as shown in Figure 3a. As shown in Figure 3b, when the current direction of the excitation coils is opposite, the excitation transducer can generate the T(0,1) mode guided wave in the pipe. For receiving the T(0,1) mode guided wave, the induced voltage of the coil should subtract the other voltage. As shown in Figure 3c, when the current direction of the excitation coils is the same, the excitation transducer can generate the T(1,1) mode guided wave in the pipe. For receiving the T(1,1) mode guided wave, the induced voltage of the coils should add up. In general, to generate T(n,1) mode guided waves, the circumference needs to be divided into 2n equal parts. The number of the permanent magnets was 2n, and the polarization of the adjacent permanent magnets was opposite. The number of the meander-line coils was 2n, and the direction of current of all the meander-line coils were the same. For different diameter pipes, it is necessary to design corresponding coils and select suitable magnets according to the circumference of the pipe. In addition, due to the magnetization of the magnet, the magnetization of the pipe under different magnet spacing conditions must be considered at the same time to ensure the coupling efficiency of the transducer.

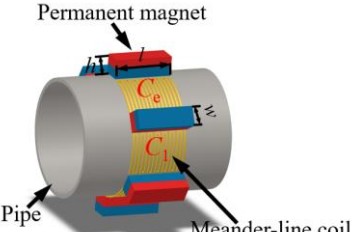

**Figure 2.** Schematic diagram of the structure of the non-contact torsional mode guided wave transducer in the steel pipe. (*l*, *w*, and *h* represent the length, width, and height of the permanent magnet, respectively).

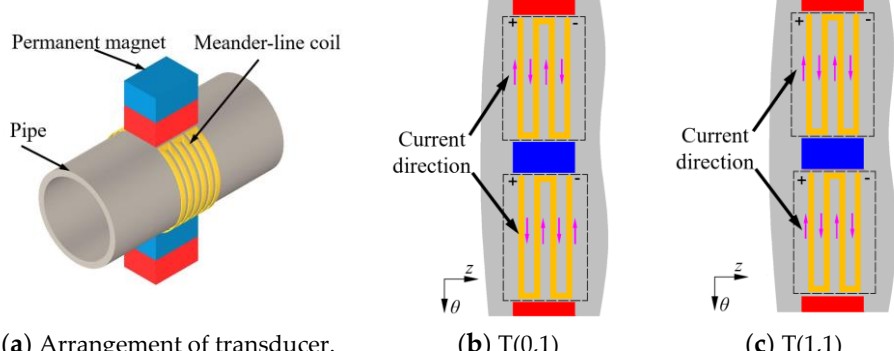

(**a**) Arrangement of transducer.　(**b**) T(0,1)　(**c**) T(1,1)

**Figure 3.** Arrangement of permanent magnets and meander-line coils of the transducer for T(0,1) and T(1,1) mode guided waves. For generating T(0,1), the current direction of the coils is opposite; for receiving T(0,1), the induced voltage of the coils should subtract the other. For generating T(1,1), the current direction of the coils is the same; for receiving T(1,1), the induced voltage of the coils should add up.

### 3.2. The Key Parameters of the Transducer

Generating the non-contact torsional mode guided wave in the steel pipe based on the Wiedemann effect is mainly realized by permanent magnets and meander-line coils. Therefore, the structural parameters of the transducer mainly include the size of the permanent magnet and the lift-off distance of the coil. This section mainly studies the influence of key structural parameters of the transducer, which are composed of two permanent magnets and meander-line coils by experiments. The influence of each key structural parameter on the efficiency of the magnetoacoustic transducer can be divided into the influence on the actuation and receiving of the T(0,1) guided wave transducer and the influence on excitation and actuation of the T(1,1) guided wave transducer.

The specifications of the steel pipe used in the experiment were outer diameter 73 mm, wall thickness 5 mm, and length 2895 mm. The dispersion curves are shown in Figure 4. The specifications of the permanent magnet used in the experiment were brand N52, length 40 mm, width 30 mm, and height 40 mm. The meander-line coil used in the experiment was made of enameled wire, and its design frequency was 250 kHz. The space period was equal to the wavelength of the T(0,1) mode guided wave, both were 12.9 mm, and each width was 4.4 mm, as shown in Figure 5a. The wire diameter of the excitation coil was 0.49 mm, and the wire diameter of the receiving coil was 0.21 mm. The total length of the coil was 86.2 mm and the total width was 36.7 mm. The circumferential section length was 71 mm and the coverage angle was about 111.45°. The transducer layout diagram is shown in Figure 5b.

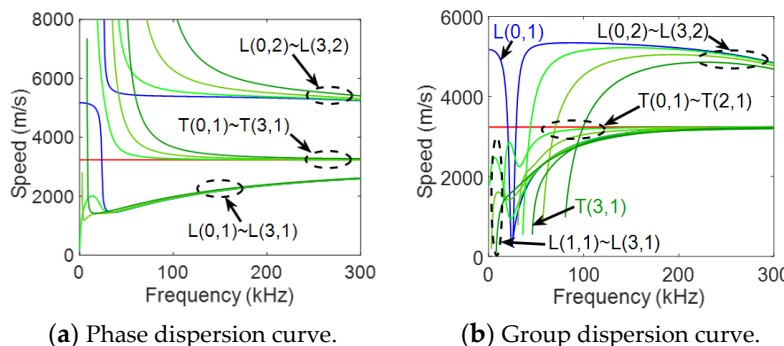

(**a**) Phase dispersion curve.　(**b**) Group dispersion curve.

**Figure 4.** The dispersion curve of the pipe (OD 73 mm, WT 5 mm, Carbon steel pipe).

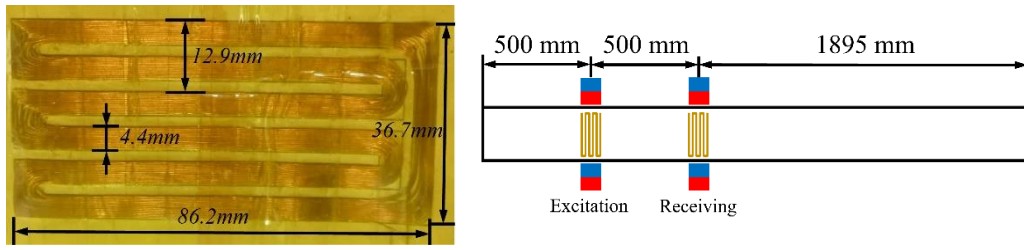

(**a**) The meander-line coil.  (**b**) Schematic diagram of transducer layout.

**Figure 5.** Experimental layout for studying the key parameters of the transducer.

A guided wave testing instrument was employed, and the structure of the instrument is shown in Figure 6. The excitation signal used in the experiment was a 4-period 250 kHz sinusoidal signal modulated by a Hanning window. The peak amplitude of the excitation signal was about 500 V. The magnification of the receiving circuit was about 2000 times. The pass frequency of the bandpass filter was 100–500 kHz, and the sampling rate was 5 MHz. In order to reduce the noise, each signal was repeatedly sampled 200 times to obtain the average. During the experiment, the size and lift-off of the permanent magnet and the lift-off of the coil were changed, and the single-factor method was used to study the influence of each key structural parameter on the guided wave signal. T(0,1) and T(1,1) mode guided wave transducers are formed by changing the connected sequence of the coil. By exchanging excitation and receiving transducers, the influence of each key structural parameter on the excitation and receiving of the transducer was studied, respectively. A typic signal obtained from the experiments is shown in Figure 7. The first signal in Figure 7 is the initial pulse, which is an electromagnetic wave that travels at the speed of light. The second signal is the first passing signal, which is the elastic wave from the actuator to the receiver in the pipe. The arrive time of the signal was about 0.1567 ms and the amplitude was about 9.9646 V. Based on the distance between the actuator and the receiver, the calculated wave speed was about 3191 m/s. The third signal is the left end echo. The fourth signal is the right end echo. The peak-to-peak value of the first passing signal in the signal was used to represent the influence of each key structural parameter on the guided wave signal, and the experimental results were normalized.

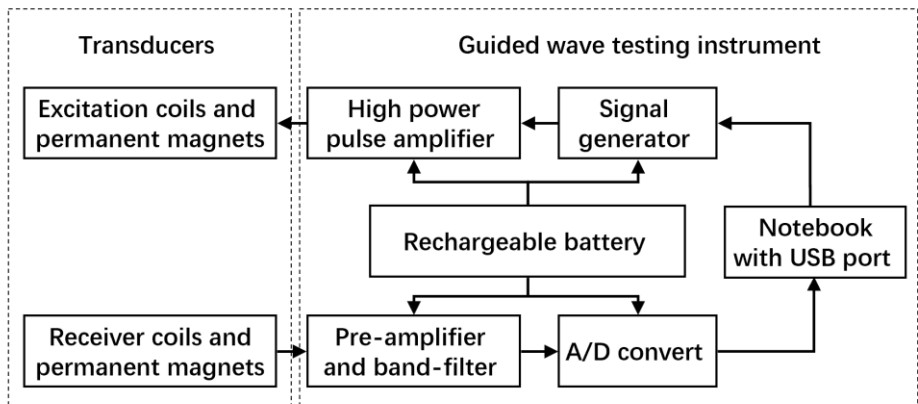

**Figure 6.** The structure of the guided wave testing instrument.

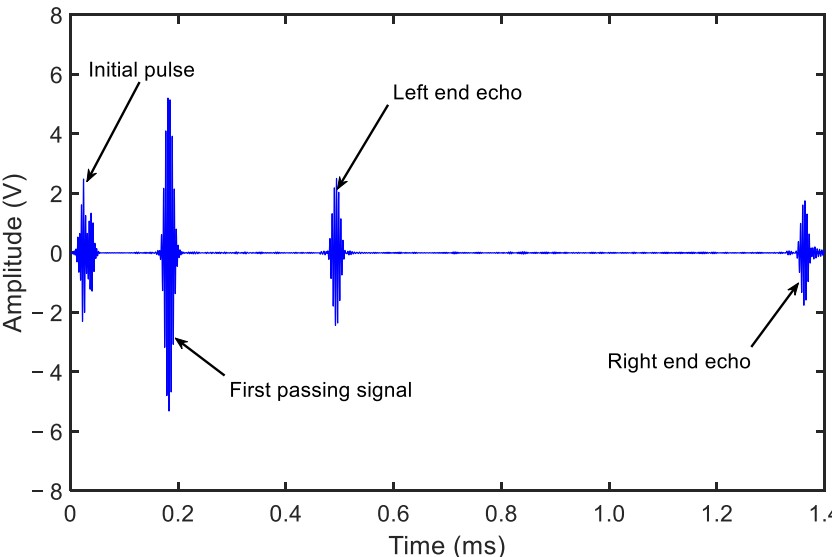

**Figure 7.** The data obtained from experimental layout shown in Figure 3b using T(0,1) mode.

### 3.2.1. The Effect of Permanent Magnet Height

In order to study the effect of permanent magnet height on the excitation and receiving sides of T(0,1) and T(1,1) guided wave transducer, the excitation (receiving) transducer was 40 mm long, 30 mm wide, and 10 mm high. The permanent magnet of N52 was gradually spliced to 60 mm along the height direction. The receiving (excitation) transducer used a permanent magnet with a length of 40 mm, a width of 30 mm, and a height of 60 mm. The magnet was directly installed on the surface of the steel pipe. The experimental results are shown in Figure 8.

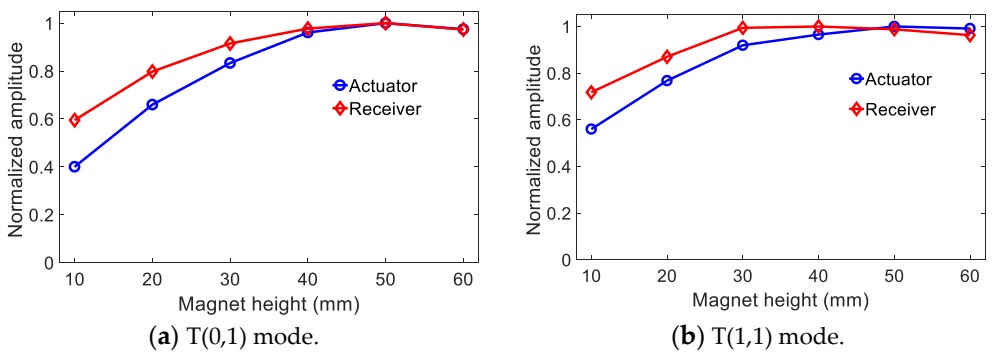

**Figure 8.** The effect of the height of the permanent magnet on the efficiency of the transducer.

The influence of the height of the permanent magnet on the excitation and receiving ends of the T(0,1) and T(1,1) modal guided wave transducers is similar from Figure 8. As the height of the permanent magnet increased, the guided wave signal gradually increased, and the increasing amplitude gradually slowed down. When the height of the permanent magnet was too high, the guided wave signal decreased slightly. As the height of the permanent magnet increased, the magnetic attraction between the transducer and the steel pipe also increased. The permanent magnet that was too high was not conducive to the disassembly or assembly process of the transducer. In addition, the increase in the height of the permanent magnet also increased the overall size of the transducer. It was necessary to select a suitable permanent magnet height so that the transducer had a suitable overall size and magnetic attraction under the premise of meeting the energy conversion efficiency of the transducer. Based on the results shown in Figure 8, the height of the permanent magnet in the transducer was selected as 40 mm.

### 3.2.2. Influence of Permanent Magnet Width

In order to explore the influence of the width of the permanent magnet on the actuator and receiver of the T(0, 1) and T(1, 1) guided wave transducers, the magnet on both actuator and receiver was 40 mm long, 60 mm high. The width of the permanent magnets were 10 mm, 15 mm, 20 mm, 30 mm. When we studied the influence of permanent magnet width on the actuator, the width of the permanent magnet on the receiver was fixed on 30 mm. When we studied the influence of permanent magnet width on the receiver, the width of the permanent magnet on the actuator was fixed on 30 mm. The coil was directly setup on the surface of the pipe. The experimental results are shown in Figure 9.

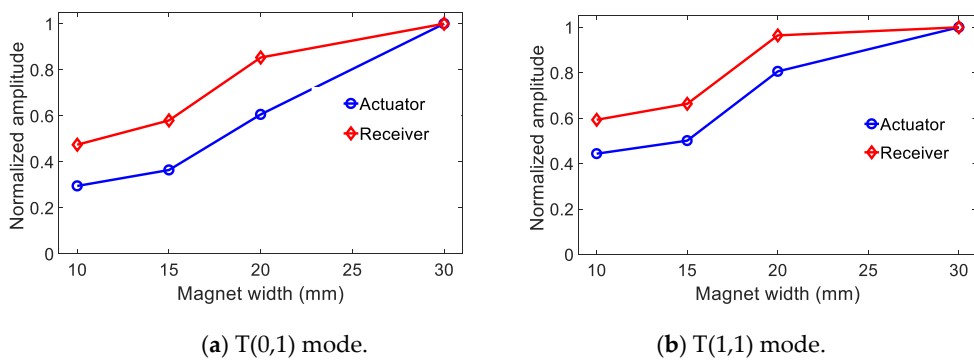

(**a**) T(0,1) mode.　　　　　　　　　　　　(**b**) T(1,1) mode.

**Figure 9.** The effect of the width of the permanent magnet on the efficiency of the transducer.

As the width of the permanent magnet increased, the amplitude of the guided wave signal gradually increased and then slowed down, as shown in Figure 9. As the width of the permanent magnet increased, the circumferential coverage angle of the folding coil decreased, and the magnetic attraction between the transducer and the steel pipe also increased. Permanent magnets were too wide to set up the transducer. In addition, the increase in the width of the permanent magnet also increased the overall size of the transducer. Combined with the experimental results in Figure 9, the width of the permanent magnet in the transducer was set to 30 mm.

### 3.2.3. The Effect of Permanent Magnet Length

In order to explore the influence of the permanent magnet length of the guided wave transducer, the magnet length of the excitation transducer was set to 5 mm, 10 mm, 15 mm, 20 mm, 30 mm, 40 mm, 60 mm, 80 mm, 120 mm, and 160 mm. The receiving transducer used a permanent magnet with a length of 40 mm, a width of 30 mm, and a height of 60 mm, with the grade N52. The permanent magnet lift-off of the excitation and receiving transducer and the coil were both set to 0 mm, and the experimental results are shown in Figure 10.

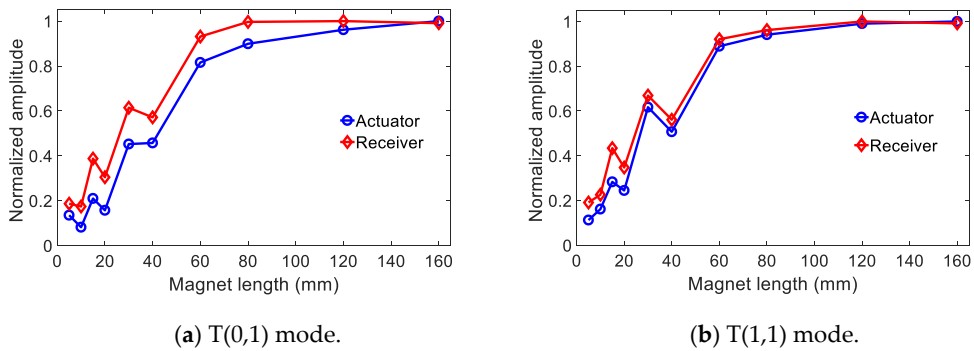

(**a**) T(0,1) mode.　　　　　　　　　　　　(**b**) T(1,1) mode.

**Figure 10.** The effect of the length of the permanent magnet on the efficiency of the transducer.

As the length of the permanent magnet increased, the amplitude of the guided wave signal gradually increased and then tended to saturate. As the length of the permanent magnet increased, the axial coverable length of the coil increased, and the magnetic attraction between the transducer and the steel pipe also increased. A permanent magnet that is too long is not conducive to the disassembly and assembly process of the transducer. In addition, the increase in the length of the permanent magnet also increased the overall size of the transducer. Combined with the experimental results in Figure 10, the length of the permanent magnet in the transducer was set to 80 mm.

### 3.2.4. The Lift-Off Effect of Permanent Magnet

Both excitation and receiving transducers used permanent magnets with a length of 40 mm, a width of 30 mm, and a height of 60 mm, with the grade N52. The lift-off distance of coils of the excitation and receiving transducers were both set to 0 mm. We used a polyurethane gasket with a thickness of about 1.4 mm to lift off the permanent magnet of the transducer. When we studied the lift-off effect of the permanent magnet on the actuator, the lift-off of the permanent magnet on the receiver was 0 mm. When we studied the lift-off effect of the permanent magnet on the receiver, the lift-off of the permanent magnet on the actuator was 0 mm. The experimental results are shown in Figure 11.

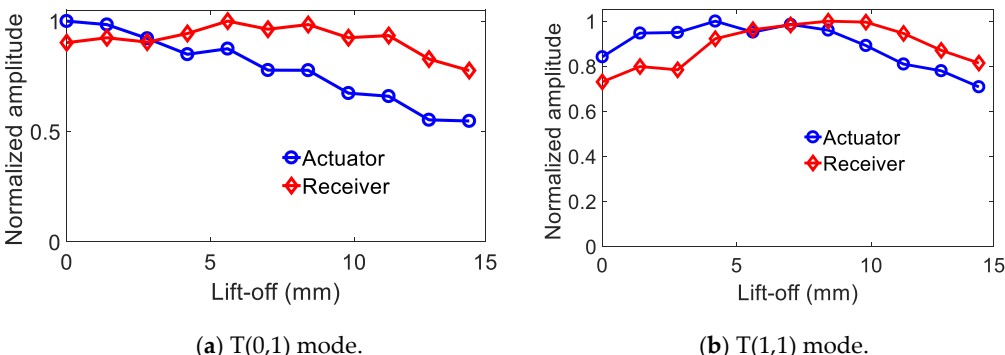

(**a**) T(0,1) mode.　　　　　　　　(**b**) T(1,1) mode.

**Figure 11.** The effect of the lift-off of the permanent magnet on the efficiency of the transducer.

As the lift-off of the permanent magnet increasef, the guided wave signal decreased slowly. To receiving the T(0,1) and T(1,1) guided wave transducer, the amplitude of the guided wave signal generally increased and then decreased. The results show that the permanent magnet lift-off had no obvious influence on the guided wave signal and proper lift-off was beneficial to the guided wave signal. As the lift-off of the permanent magnet increased, the magnetic attraction between the transducer and the steel pipe weakened, which is beneficial to the disassembly and assembly process of the transducer. In addition, the increase in permanent magnet lift-off also increased the overall size of the transducer, and too low permanent magnet lift-off made the permanent magnet housing thinner in the main force direction. Combining the experimental results in Figure 11, and considering the phenomenon that the magnetic attraction between the permanent magnet and the steel pipe during the experiment leads to the thinning of the polyurethane gasket, the lift-off of the permanent magnet in the transducer was set to 3.5 mm.

### 3.2.5. The Effect of the Lift-Off of the Coil

Both excitation and receiving transducers used permanent magnets with a length of 40 mm, a width of 30 mm, and a height of 60 mm, with the grade N52. The lift-off of the permanent magnets of the excitation and receiving transducers were both set to 0 mm. We used a polyurethane gasket with a thickness of about 1.4 mm to lift off coil of the transducer. When we studied the lift-off effect of the coil on the actuator, the lift-off of the coil on the receiver was 0 mm. When we studied the lift-off effect of the coil on the receiver, the lift-off of the coil on the actuator was 0 mm. The experimental results are shown in Figure 12.

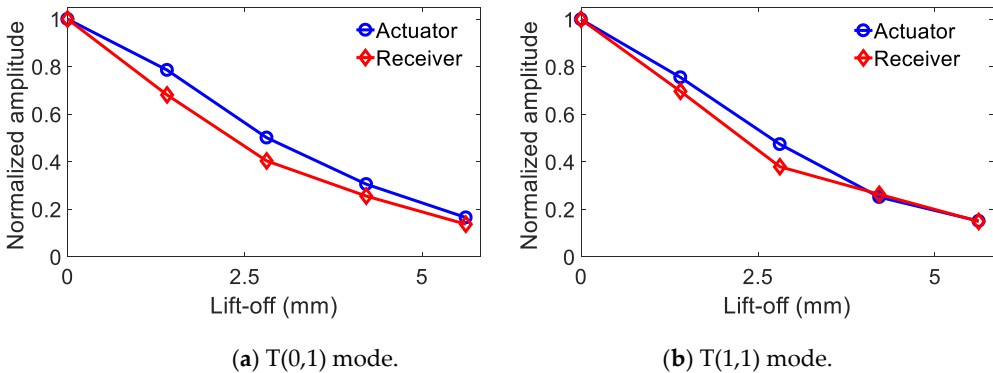

(**a**) T(0,1) mode.  (**b**) T(1,1) mode.

**Figure 12.** The effect of the lift-off of coil on the efficiency of the transducer.

As the coil lift-off increased, the amplitude of the guided wave signal decayed fast. When a layer of polyurethane gasket was placed between the coil of the transducer and the steel pipe, the guided wave signal attenuated to 80%. Therefore, the coil lift-off in the transducer cannot be too large, but too low coil lift-off makes the coil housing too thin and unable to be processed. Combining the experimental results in Figure 12, and considering that the 3D printing nylon material is deformed when the thickness is less than 0.8 mm, the lift-off of the coil in the transducer was set to 1 mm.

### 3.3. The Prototype of the Sensor

The main function of the magnetization module designed in this section is to realize the functions of fixing the permanent magnet, moving the transducer along the axial and circumferential direction of the steel pipe and connecting the coil module. Combining the previous analysis, two magnetization modules were designed, as shown in Figure 13.

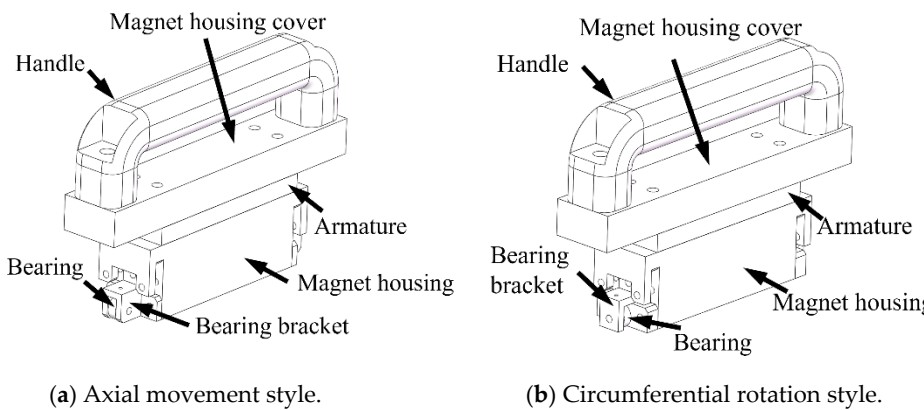

(**a**) Axial movement style.  (**b**) Circumferential rotation style.

**Figure 13.** Three-dimensional model of the magnetization module.

Both magnetization modules use bearings to realize the movement of the transducer along the surface of the steel pipe. The magnetization module shown in Figure 13a can make the transducer move in the axial direction of the steel pipe; the magnetization module shown in Figure 13b can make the transducer rotate in the circumferential direction of the steel pipe. The function of the handle is to facilitate carrying and disassembling the transducer. The function of the magnet housing cover is to connect and fix the magnet housing, the armature, and the handle and to prevent the permanent magnet and the armature from slipping out. The function of the armature is to assist in the splicing and installation of the permanent magnet and to improve the force of the magnet housing between the permanent magnet and the steel pipe. Two permanent magnets with a length of 40 mm, a width of 30 mm, and a height of 40 mm were used in the assembly process of the transducer. The permanent magnets of grade N52 are spliced along the length direction. Because the repulsive force between the same magnetic poles was too large, an armature

was needed for auxiliary installation. The function of the bearing bracket is to install the bearing to realize the movement of the transducer. In addition, a threaded through hole was opened on the bearing bracket, and a set screw can be installed to fix the transducer on the steel pipe.

The function of the magnet housing is to install the permanent magnet, the bearing bracket, and connect the coil module. The structure of the magnet housing is shown in Figure 14. In order to facilitate the installation of permanent magnets, the length, width, and depth of the inner cavity of the magnet housing were 82 mm, 31 mm, and 40 mm. The bottom surface of the magnet housing was the main force-bearing surface, with a thickness of 2.5 mm. The minimum distance between the magnet housing and the surface of the steel pipe was 1 mm. Excessive thickness of the two sides of the magnet housing in the width direction of the permanent magnet reduces the circumferential coverage angle of the folding coil, while too thin thickness of the two sides reduces the strength of the magnet housing, so the thickness was set to 2 mm.

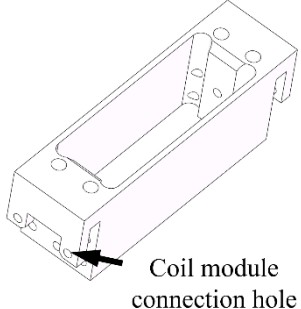

**Figure 14.** Three-dimensional model of magnet housing.

The main function of the coil module is to realize the functions of fixing the coil and BNC connector, coil routing, and connecting the magnetization module. Combining the previous analysis, a square module was designed for the single-hinge connection scheme, where the magnetized module was disconnected, and its internal structure is shown in Figure 15.

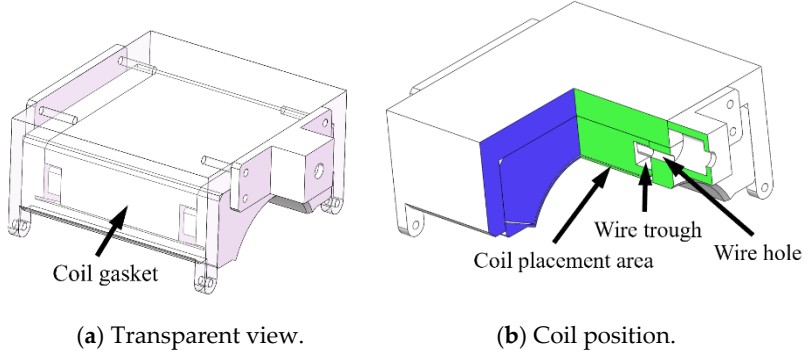

(**a**) Transparent view.　　　　(**b**) Coil position.

**Figure 15.** The internal structure of the coil module.

The double-layer coil in the module is shown in Figure 16. Its design frequency was 250 kHz, the space period was 12.932 mm, and each fold width was 4.4 mm, with a total of 10 folds. The line width of the excitation coil was 0.8 mm, with five turns per single layer, and the line width of the receiving coil was 0.275 mm, with nine turns per single layer. The total length of the turn-back coil was 110 mm, the total width was 62.594 mm, the length of the circumferential section was 84 mm, and the circumferential coverage angle of the circumferential section was about 127°.

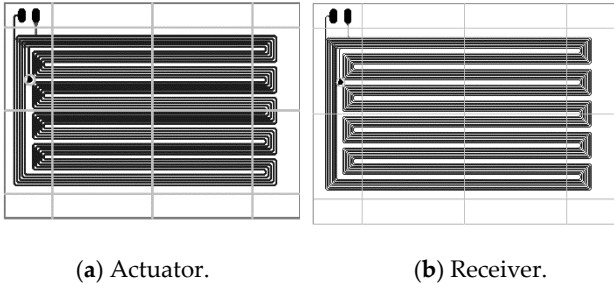

(**a**) Actuator.            (**b**) Receiver.

**Figure 16.** FPCB meander-line coil.

The magnetization module and coil module was designed and installed on a 73 mm outer diameter steel pipe. The overall installation structure is shown in Figure 17. The coil module and the magnet module were connected by bolts. After the transducer was installed on the steel pipe, the minimum distance between the magnet housing and the steel pipe surface was 1 mm. The minimum distance between the coil housing and the steel pipe surface was 0.2 mm. The transducer was in contact with the surface of the steel pipe through a bearing, which realized the axial movement and circumferential rotation of the transducer.

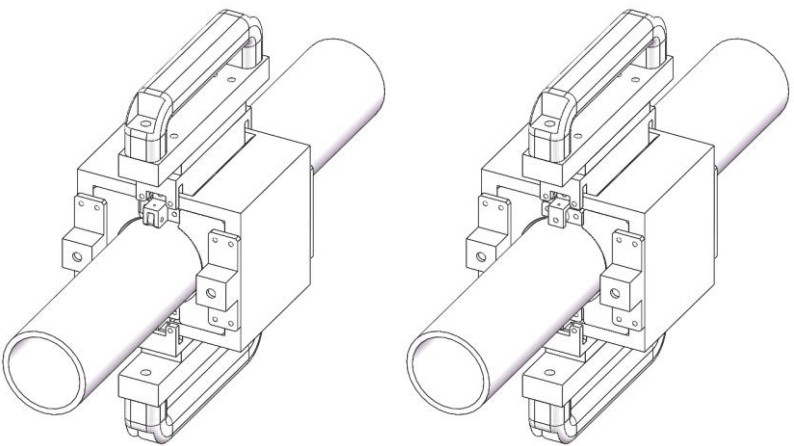

(**a**) Axial movement style.            (**b**) Circumferential rotation style.

**Figure 17.** Assembly drawing of the overall structure of the transducer.

In order to reduce the influence of the magnetization module on the static magnetic field formed by the permanent magnet in the steel pipe, except for the armature, the other parts were made of high-strength and non-magnetic aluminum alloy and the material of the armature was selected from low carbon steel. The module is shown in Figure 18.

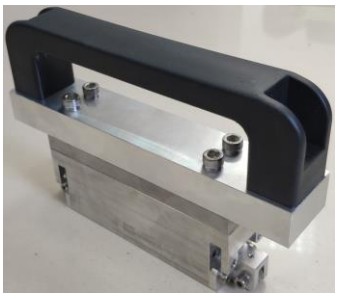
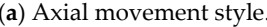
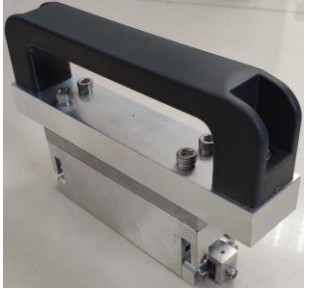

(**a**) Axial movement style.            (**b**) Circumferential rotation style.

**Figure 18.** Photo of the magnetized module.

In order to reduce the influence of the coil module on the static magnetic field formed by the permanent magnet in the steel pipe, the material of the coil housing cover and the connecting plate was made of high-strength and non-magnetic aluminum alloy and the material of the coil backing plate was nylon, which was made and assembled. The coil module is shown in Figure 19.

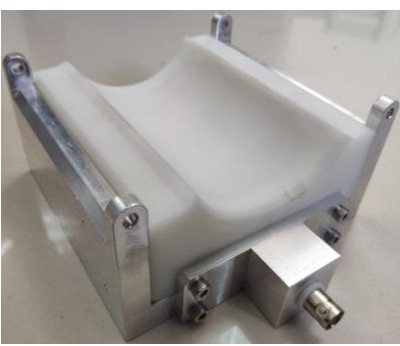

**Figure 19.** Photo of coil module.

We assembled the manufactured and assembled the magnetization module and coil module into a transducer, as shown in Figure 20. In addition, the bolts and gaskets used in the transducer were all made of 304 stainless steel to minimize the impact on the static magnetic field formed by the permanent magnet in the steel pipe.

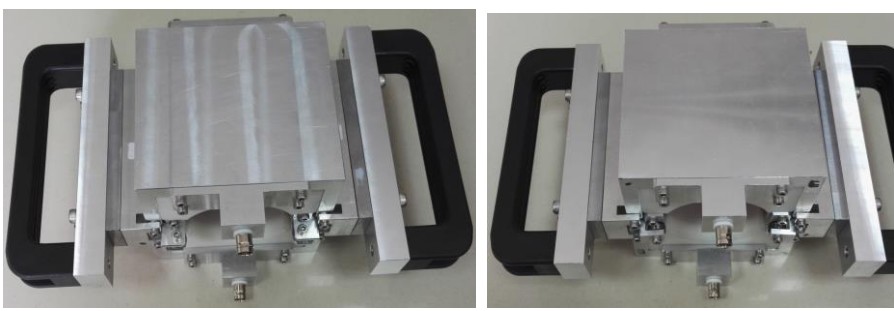

(**a**) Axial movement style.          (**b**) Circumferential rotation style.

**Figure 20.** Photo of non-contact guided wave transducer based on the Wiedemann effect.

## 4. Experiments of Validation

The experimental platform is shown in Figure 21, which includes a guided wave testing instrument with PC, an actuator, and a receiver. A steel pipe with a 73 mm diameter, 5 mm wall thickness, 2895 mm length was employed as the specimen.

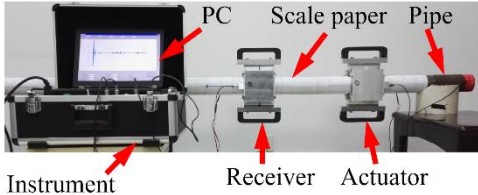

**Figure 21.** The photo of the experimental platform.

### 4.1. Defect Detection Ability and Effective Detection Range of the Transducer Using T (0,1)

To study the defect detection ability, a notch defect was machined on the steel pipe. The cross-sectional area loss rate of defect increased from 1% (the depth of the notch was about 1 mm) to 3% (the depth of the notch was about 2 mm) in 0.5% steps. The distribution

of the defect and the placement of the transducer in the pipe are shown in Figure 22. To study the defect detection ability of the transducer, only the T(0,1) mode was employed because of its axial symmetry. Other parameters used in the experiment were consistent with Section 3.2. The data obtained from 1% cross-sectional area loss notch is shown in Figure 23.

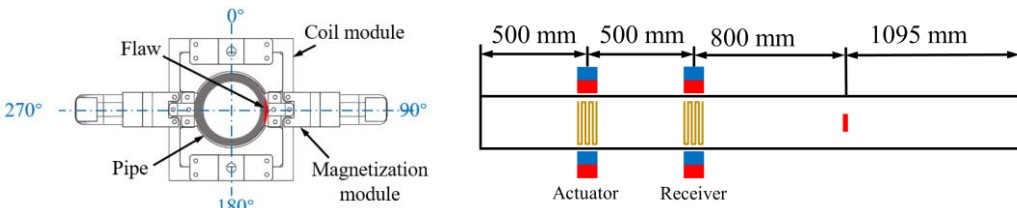

(**a**) The distribution of the defect.

(**b**) The location of the defect and the placement of the transducer.

**Figure 22.** The experiment setup for study the defect detection ability of the transducer.

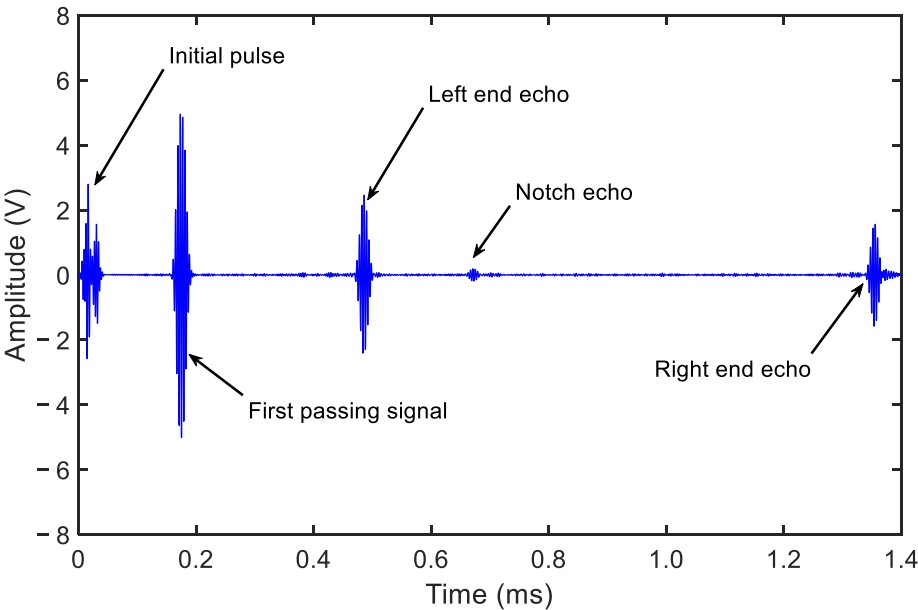

**Figure 23.** The data obtained from 1% cross-sectional area loss notch using T(0,1) mode.

The fourth signal is the notch echo and the Vpp was about 0.3788 V. The relation between the cross-sectional area loss of the notch and the Vpp of the notch echo is shown in Figure 24. There was almost a linear relation between the cross-sectional area loss of the notch and the Vpp of the notch echo.

To calculate the effective detection range, the left end echo and the right echo were employed to calculate the attenuation coefficient. The propagation distance of the left end echo was 1500 mm, and the amplitude was about 4.8658 V. The propagation distance of the right end echo was 4290 mm, and the amplitude was about 3.5385 V. The attenuation coefficient was about −0.9879 dB/m. The amplitude of the noise was about 0.0759 V. Here, we define that, if the signal-to-noise ratio is greater than 2, the defect signal is considered to be identifiable. Therefore, the effective detection range of 1% cross-sectional area loss notch was about 4.8200 m and 3% notch was about 9.8707 m.

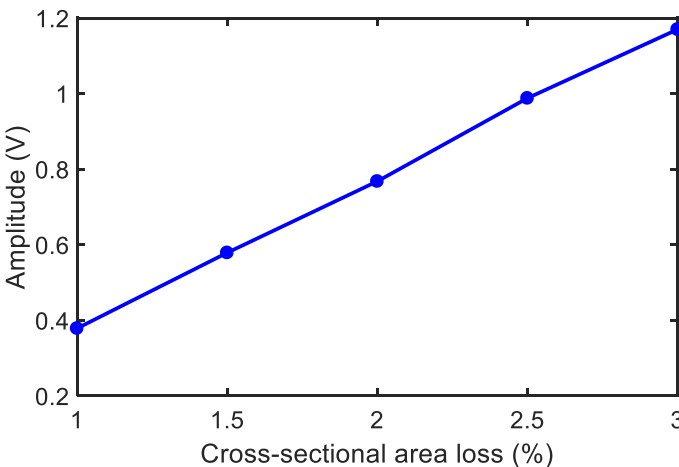

**Figure 24.** The relation between the cross-sectional area loss of the notch and the Vpp of the notch echo.

### 4.2. Locating Defect Using T(0,1) and T(1,1)

In the experimental platform shown in Figure 21, the function of the scale paper on the outer surface of the steel pipe was to determine the defect of the transducer and the groove circumferential position. The axial position of the transducer and the notch defect is shown in Figure 25a. The depth of the notch defect was about 2 mm, and the cross-sectional area loss was about 3%. The initial installation position of the transducer in the circumferential direction and the circumferential position of the center of the groove defect are shown in Figure 25b.

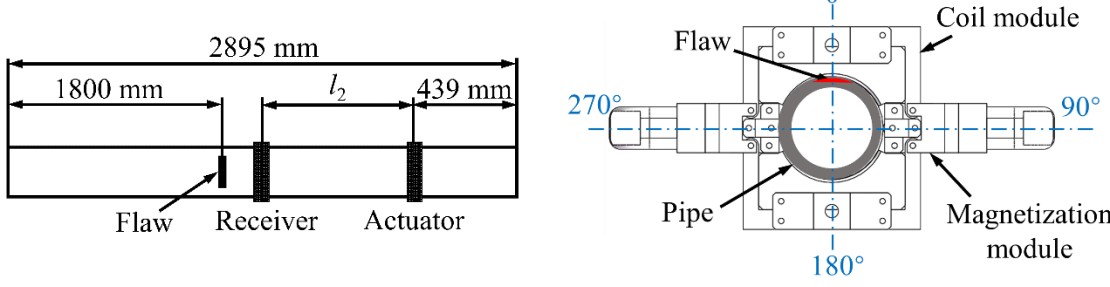

(**a**) Axial installation position of transducer.
($l_2$ = 440 or 540 mm)

(**b**) Circumferential installation position of transducer.

**Figure 25.** Schematic diagram of transducer installation position to locate the defect.

The defect detection signal obtained by axially moving the receiving transducer is shown in Figure 26. Since T(0,1) and T(1,1) guided waves have non-dispersive characteristics at 250 kHz, the two mode guided waves can be used to locate defects in the axial direction. When the transducer spacing was 440 mm, the receiving time of the defect echo in the T(0,1) and T(1,1) modal guided wave detection signal was 269.6 μs and 270.4 μs, respectively. Combined with the dispersion of the steel pipe curve, the propagation distance of the defect echo was calculated to be 871.62 mm and 872.58 mm, respectively. Then the receiving transducer was moved axially to increase the transducer spacing to 540 mm, and it can be clearly observed that the defect echo in the detection signal arrived earlier. Therefore, it can be judged that the defect was located at a position 215.81 mm (T(0,1)) or 216.29 mm (T(1,1)) from the left side of the receiving transducer, which was close to the actual axial position of the defect. The relative errors of the axial positioning of the defects were 0.09% (T(0,1)) and 0.13% (T(1,1)), respectively. In summary, the manufactured non-

contact guided wave transducer uses T(0,1) or T(1,1) guided waves to achieve non-contact detection of defective steel pipes and axial defects position.

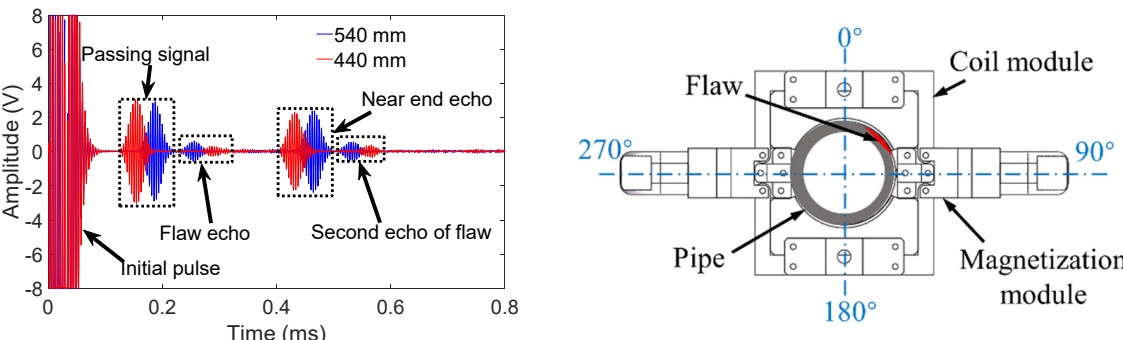

(**a**) T(0,1) when the central of the flaw locates 45°.

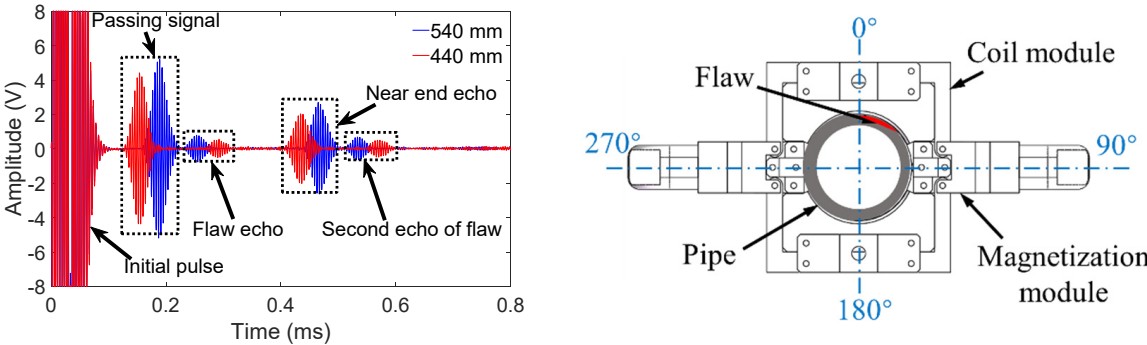

(**b**) T(1,1) when the central of the flaw locates 22.5°.

**Figure 26.** Defect detection signal obtained by axially moving the receiving transducer (the distances between the actuator and the receiver were 440 mm (red) and 540 mm (blue)).

We rotated the excitation transducer along the circumference of the steel pipe with a step angle of 22.5° and then obtained the peak-to-peak value of the defect echo in the detection signal at each circumferential position of the excitation transducer and performed normalization processing. The results are shown in Figure 27.

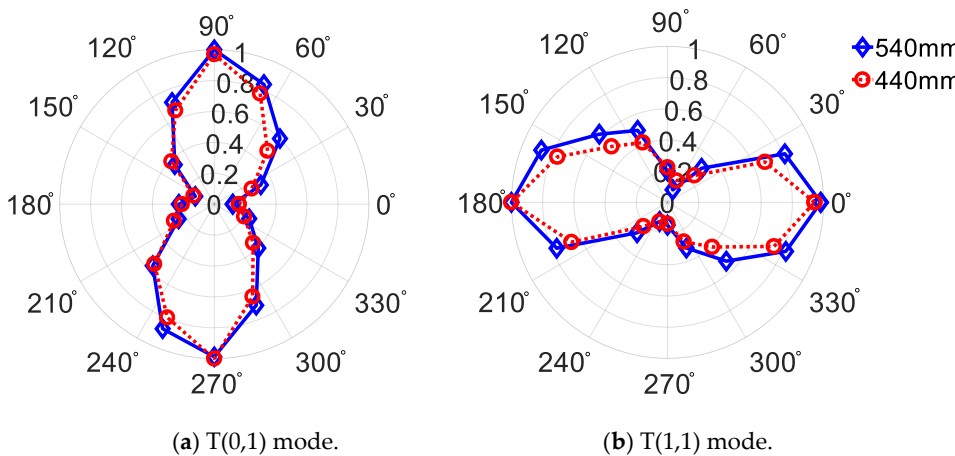

(**a**) T(0,1) mode.　　　　　　　(**b**) T(1,1) mode.

**Figure 27.** The signal obtained by the circumferential rotation excitation transducer (the distances between the actuator and the receiver were 440 mm (red) and 540 mm (blue)).

It can be seen from Figure 27 that the results obtained when the transducer distance was 540 mm and 440 mm are similar. For T(0,1), when the excitation transducer was located at 90° or 270°, the defect echo reached the maximum. When the excitation transducer was located at 0° or 180°, the defect echo was very small. For T(1,1), when the excitation transducer was located at 0° or 180°, the defect echo reached the maximum. When the excitation transducer was located at 90° or 270°, the defect echo was very small. When the designed transducer excited the T(0,1) or T(1,1) mode guided waves, it was accompanied by undesired mode guided waves that were difficult to suppress. The other undesired mode guided waves had similar non-dispersive characteristics, and its group velocity was also similar to T(0,1) and T(1,1). However, the superposition of undesired mode guided waves and T(0,1) or T(1,1) mode guided waves affected the distribution of guided wave energy in the circumferential direction of the steel pipe. Because the transducer excited the T(0,1) mode guided wave, the proportion of the undesired mode guided wave was larger. Therefore, this part of the superposition had a greater effect on the guided wave of the T(0,1) mode. The maximum value of the guided wave energy in the circumferential direction of the steel pipe was shifted to a circumferential position 90° away from the center of the coil. Using T(0,1) and T(1,1) modal guided waves at the same time can avoid defect leakage when the transducer cannot be excited by circumferential rotation. Due to the difference in wave structure between T(0,1) and T(1,1) modes, the two modes can complement each other and provide support for the circumferential positioning of defects. In addition, the circumferential position of the center of the grooved defect was the same as the circumferential position of the excitation transducer when the peak-to-peak value of the defect echo in the T(1,1) guided wave detection signal reached the maximum. When the peak-to-peak value of the defect echo in the T(0,1) guided wave detection signal reached the maximum, the circumferential position of the excitation transducer differed by 90°. The transducer can be used to realize locating the circumferential positioning of defects.

## 5. Conclusions and Future Work

This paper provides a non-contact electromagnetic guided wave transducer based on the Wiedemann effect for pipeline inspection. The transducer is mainly composed of permanent magnets and meander line coils. The static bias magnetic field generated by the permanent magnet and the alternating magnetic field generated by the coil are perpendicular to each other. Two types of transducers with circumferential rotation and axial movement were developed. The transducer can generate and receive the T(0,1) and T(1,1) modes in the steel pipe. The axial positioning of the defect is located by moving the transducer axially. The circumferential positioning of the defect is located by rotating the transducer. Additionally, missed detection can be effectively avoided by rotating the transducer. In the future, we will develop the transducers for pipes with different diameters and study the effective detection distance of these transducers.

**Author Contributions:** Conceptualization, J.X. (Jiang Xu), G.C., J.X. (Jin Xu); methodology, J.X. (Jiang Xu), J.X. (Jin Xu) and G.C.; validation, G.C. and J.X. (Jin Xu); formal analysis, J.X. (Jiang Xu), G.C., J.X. (Jin Xu); investigation, J.X. (Jiang Xu), G.C., J.X. (Jin Xu); data curation, G.C., J.X. (Jin Xu); writing—original draft preparation, J.X. (Jin Xu) and J.X. (Jiang Xu); writing—review and editing, J.X. (Jin Xu), J.X. (Jiang Xu) and Q.Z.; visualization, J.X. (Jin Xu), J.X. (Jiang Xu) and Q.Z.; supervision, J.X. (Jiang Xu) and Q.Z.; project administration, J.X. (Jiang Xu); funding acquisition, J.X. (Jiang Xu). All authors have read and agreed to the published version of the manuscript.

**Funding:** This work was supported by the National Natural Science Foundation of China (Grant No.51575213).

**Conflicts of Interest:** The authors declare no conflict of interest.

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
