# Peer review of "Guided Wave Transducer for the Locating Defect of the Steel Pipe Based on the Weidemann Effect"

_actuators, doi:10.3390/act10120333_

Round 1
Reviewer 1 Report
This paper introduces two non-contact electromagnetic guided wave transducers (with circumferential rotation and axial movement) based on Wiedemann effect for flaw pipeline detection. The detection of this type of failure with this type of transducer are very interesting. The principle of operation and the main equations are presented. The structure designed for the transducer and its key design parameters are described below. Experimental results obtained as an effect of the modification of the dimensions of the permanent magnet are shown.
Experimental results obtained as an effect of the modification of the dimensions of the permanent magnet, also those due to the lift-off of the permanent magnet and the coil, are shown, but a previous theoretical study is not provided to justify the variations considered nor a guide is provided for design and selection of those parameters.
A detailed description is made of the prototype of the sensor built for a certain pipe with which the experimentation has been carried out. It would be desirable to establish the detection range for which it would be valid (pipe dimensions and distances to failure) and how it should be adapted to other dimensions.
Some details about the excitation and detection circuits used in this experiment would be interesting.
Author Response
Please see the attachment, thank you.

Reviewer 2 Report
Authors in this research designed a non-contact magnetostrictive torsion mode guided wave transducer based on Weidemann effect. The design is realized by permanent magnets and meander-line coils. The transducer can locate axial position and circumferential position of the defect in the steel pipe.
This research can be improved from the following aspects.
- Explain more clearly about the logic behind the transducer design
The logic behind the transducer design is introduced in the first paragraph about "3.1. The structure of the transducer", which is important for this research. The logic is straight forward but seems confusing. The authors may need to polish this paragraph for easier understanding. For example, they can introduce permanent magnets and meander-line coils separately instead of in a single sentence.
- Spend more effort to demonstrate the performance of this transducer
The authors explain in detail about how to choose the key parameters of the transducer: permanent magnet height, width, and length; lift-off effects of permanent magnet and meander-line coils. This explanation is good for readers to understand the system. However, the performance of the system is typically of more interest. In addition to the introduced experiment in "4. Experiments of validation", authors may need to vary the flaw size to show the system's ability in detecting small flaws. Also, authors may need to quantify the detection SNR for receiver and actuator at different distances to flaw. Thus, the detection range of the system can be demonstrated.
- Try to have more intuitive visualizations in Figures 17 and 18
When readers see Figures 17 and 18, they need to go to texts to find information about the flaw and the explanations why this flaw’s axial position and circumferential position can be determined by the system. In fact, the flaw can be visualized in Figures 17 and 18 so that readers can understand it by directly seeing the figures.
Author Response
Please see the attachment, thank you.

Reviewer 3 Report
The manuscript describes the concept of magnetostrictive transducers that generate torsional guided waves in pipes. The study is interesting, however, there are language issues that make it hard to read. There is also mistakes, typos, missed spaces and punctuation. Additionally, it lacks deeper explanations in the theoretical section. A throughout revision and proofreading is necessary.
In addition, I have the following comments:
-
Section 2 needs to be more in deep explained and more formalism is required. Use some features for matrices and tensors (boldface, underline, etc). A schematic figure with the static and dynamic magnetic fields would help.
What do the authors mean with Eq. (3) being ignored (lines 110 and 111)? Is this because only axial propagation (torsional modes) were considered? This is confusing and needs to be clarified in the manuscript.
-
In section 3.1, it is not clear how the selective generation of modes T(0,1) and T(1,1) or T(n1,) is performed. Is this due to the direction of injected currents into the coils (or how the coils are interconnected)? This needs to be made clear in the manuscript. Also, include in Fig. 1 a schematic representation of the static and dynamic magnetic field, as well as the generated forces for generation of T(0,1) and T(n,1) and explains, based on the modes’ profile, the mechanism for which each of these modes is predominantly generated. This is briefly commented in line 182 but needs to be made much clearer also with a schematic.
-
Section 3.2, line 160: It is stated that the transducer is composed of two magnets. This is not consistent with Fig.1. This needs to be made clear in the manuscript.
Mark all pertinent dimensions reported in the paragraph ranging from line 165 to 174 in Fig. 2(a). Also, define wc and Tc in Fig.2(a).
Which equipment is used in the experimental setup?
Shown an example of a typical received signal.
-
Section 3.2.1, line 194. What do the authors mean by “The separation is set to 0 mm”? Which separation is this?
-
Section 3.2.2, lines 213 to 220 is difficult to understand. Is everything that is related to “receiving transducer” actually referring to “receiver (resp. transmitter) transducer” so the separate effect on transmitter and receiver could be assessed? Make this clearer. The same holds for section 3.2.2, lines 233 to 238.
-
Section 3.2.4 and 3.2.5 use mm in the abscissa of Fig. 6 and 7 instead of the number of layers.
How many gaskets layer were used at last in each analysis?
-
Line 398: “Since T(0, 1) and T(1, 1) guided waves have non-dispersive characteristics at 250 kHz”: show the dispersion curve for this pipe diameter and material in order to illustrate this.
-
Fig. 18: include units in the legend for blue and red markers and the symbol for degrees in the angular axis.
-
Lines 422-437: why the defect response for T(0,1) and T(1,1) are 90 degrees rotated? Explain this in the manuscript. I suggest that a radiation diagram of the generated wave modes by the transducer be provided and explained based on its generating mechanism.
-
Line 429-431. It is suggested to use the modes T(0,1) and T(1,1) simultaneously. How could this be performed practically?
-
Line 98 “flexibility matrix” → “compliance matrix”.
Author Response
Please see the attachment, thank you.

Round 2
Reviewer 3 Report
The authors attempted to address most of the reviewer's comments, however, more attention is required to address the following:
1) For the sake of completeness I still think that the explanation reported in the response letter, regarding original comment (1) needs to be included in the manuscript, but, of course, differently from previous references. Including figures similar to Fig. 1 of the response letter, the definition of piezomagnetic coupling matrix, etc.
2) The explanation is satisfactory, but in new Fig, 2, the direction of current and forces for the two generations cases, namely mode T(0,1) and mode T(1,1), need to be included in the figure by means of some feature, as arrows; use Fig.2(a) and Fig.2(b), for instance.
3) The third comment was not answered satisfactorily:
-The dimension mentioned in the text (former lines 165 to 174, now lines 173 to 182) are not explicitly highlighted in former Fig. 2 (now Fig.3). This helps the readers to follow the experimental setup.
-A schematic of the experimental setup was provided in the response letter, but not in the manuscript. This is relevant information and needs to be included in the manuscript. Also, in addition to including the schematic, the setup needs to be explained in more detail in the manuscript.
-The same holds for the example of a typical received signal: include it in the manuscript and further comment on its meaning.
7) Similar to previous comments, the dispersion curves and some comments need to be included in the manuscript (not only in the response letter) so the reader can appreciate its low/non-dispersive nature.
9)Revise the new content of text from line 484 (there are repetitions).
10) It is still not clear to me how the transducer can be set up to generate both modes simultaneously. Clearly explain this in the manuscripts, for instance stating the current direction/magnets, etc.
